

# A Mathematical Framework for Crisis Spatial Crowdsourcing Services

Hooshang Eivazy , Mohammad Reza Malek,

Department of GIS, Geodesy and Geomatic Engineering, K.N Toosi university of Technology, Tehran, 1969764499, Iran
*Correspondence to*: Mohammad Reza Malek (mrmalek@kntu.ac.ir)

**Abstract.** Almost all crises or disasters have vast consequences in both spatial and temporal scale which are difficult to manage because of the size of hazards and their complexities. To conquer the vast size of consequences, it is possible to deploy potential of crowds but there are few developed tools to come over the complexity of required emergency processes. Dividing complex process into atomic crowdsourcing services would be a proper solution to use potential of crowds. We

have innovated these services as atomic reply to spatial urgent requests including an informative, activity and confirmative segment. Proper composition of such atomic services in both spatial and temporal components would be able to manage sophisticated crisis fields. Administration of such services requires a robust and operational framework. In this paper we have designed a crisis crowdsourcing mathematical framework which is constructed upon vector space elements and some meaningful eligible operations among them. The framework inherits tensor and differential geometry specifications. Vectors

simulate crisis environment main objects in a multidimensional space. Likewise operations coincide possible actions among crisis objects.

Proposed Framework introduces tools to compose and arrange atomic crowdsourcing services in optimum mode to conquer complex aiding process. Framework offers a mathematical intelligible documentation method too which is able to record process details as well as objects' status in space. To evaluate our framework we simulate different scenarios in a commercial

simulation software and in our proposed framework. Defined scenarios included real field crisis reports, multi-agents interactive challenges for aiding and inquiry research forms for overall evaluation. Comparing results revealed considerable coincidence that could be evaluated as framework assurance.


## 1. Introduction

Conquering natural hazards consequences are usually difficult because of their huge extent in spatial and temporal scale. Early aiding systems for emergency responses to natural hazards is usually suffered from vast extent of disaster, field recognition, few number of active rescuers, dynamic and effective management and etc. Crowdsourcing potentials would be

proper solution to cover mentioned deficiencies.

Crowd sources in comparison with volunteer resources, include more people with less expert and interests. Although it doesn't support high standards of human resources for emergency services, but it fulfils other requirements such as




abundance and spatial distribution. Nevertheless effective management of many uncoordinated people with few interests and potentials would be so difficult. To overcome this problem, a new type of crowdsourcing services was introduced. This IT based service, has been designed to guide the early emergency system rescuers in form of crowdsourcing resources through the disaster or crisis filed.

Spatial Crowdsourcing services in basic and very simple form, digest complicated and long aiding mission of a single individual injured from a critical situation to safe and stable one. Services are classified to recognition, rubbing, first medical aiding, carrying, feeding, sheltering and etc. Each service at its primary segment guides a rescuer according to his capabilities and tell him where to go. Then guides him what to do and finally takes a confirmation whether the mission has been completely done, but the problem is over managing issues.

Managing many rescuers via crowdsourcing services over field at different situations with several skills in central form is as difficult as finding best arrangement of crowdsourcing services to aid individuals or critical situations. So the problem is over the arrangement and allocation of services.

There are a lot of researches regarding allocation and arrangement of tasks in project management issues but few researches has ended in an operational solution. Although software such as Float (Float , 2018) and Guru (GURU, 2018) present a

plenty of tools to scheduling tasks and operations (Ben, 2018), but there are few functionalities for auto management. In the other words, they are man-based scheduling software which offer a rich set of task scheduling tools.

Research affairs have covered different approaches such as simulation, genetic algorithms, expert systems and fuzzy theory (Mota et al., 2009), (Daher and Almeida, 2010), (Kang et al., 2011). They determine a fit between candidates' set of skills and skills which are required for a given tasks. But they don't meet the all tasks that are required for any given injured during

rescuing mission. In addition they don't present a strong framework for investigation on crowdsourcing service administration.

As these services are new in research areas, powerful frameworks are needed to investigate administration of critical issues via crowd sources. Generating measurement criteria, combining different resources, interaction among objects and many other objects are among the issues should be tested in suitable environment. Such environment should support simple strong

computation and easy further develops.



In this paper we propose a mathematical framework to manage spatial crowdsourcing services over field with mentioned specifications. It presents robust and flexible mathematical framework which lets for vast investigations over crowdsourcing services and offer a rich set of math tools for various object manipulation. In following sections, first we consider crowdsourcing and traditional frameworks including mathematical ones in section 2. Then we express structure of our proposed framework and its element details in section 3. In section 4 we define a scenario to test our framework and finally in the last section, we conclude the positive and negative points regarding our framework and try to interpret its basic capabilities.

## 2. Related works

Primary definitions of crowdsource, mostly stressed on the large group of people and open call. Such definition latterly was further developed by nearly concepts such as human computation, social computing and collective intelligence (Quinn and Bederson, 2011).

Crowd sourced resources have great potential that in recent years (Moffit and Dover, 2010) absorbed many investigators in different field specially marketing (Howe, 2009). Data mining and software engineering often utilize the crowdsourcing potential (Gansky, 2012). It has also entered to marketing and social science (Dawson and Bynghall, 2012). Likewise it has freed the location limits from LBS services (Zhao and Han, 2016). Although most researchers significantly have used the shared subjective strength of crowd resources, it is possible to deploy other aspects of crowd resources (Kazemi and Shahabi, 2012). Although crowdsourcing would be subjected to many new investigations but there is not enough fundamental and conceptual tools to further development.

In English word, a framework is an essential supporting structure of a building, vehicle, or object (Oxford Dictionary, 2016). In general, a framework is a real or conceptual structure intended to serve as a support or guide for the building of something that expands the structure into something useful.

In computer programming, a software framework is an abstraction in which software providing generic functionality can be selectively changed by additional user-written code, thus providing application-specific software. A software framework is a universal, reusable software environment that provides particular functionality as part of a larger software platform to facilitate development of software applications, products and solutions. Software frameworks may include support programs, compilers, code libraries, tool sets, and application programming interfaces (APIs) that bring together all the different components to enable development of a project or system (Riehle and Dirk, 2000).

Although software frameworks have simplified simulation process in different engineering fields, but there are some deficiencies to deploy them. Interacting with their GUIs (Tamir et al., 2015), losing control of details, reliant on framework to understand dependencies, hidden encapsulated running code to developer, broken-ability to framework updates (Argon Design, 2015).



Computer frameworks would be further developed with potential in crowds. This has created an interesting type of framework which is based on crowdsourcing. This framework has been used to improve VGI (Karam and Melchiori, 2013). Generalization of main idea behind such frameworks, facilitates construction of a friendly and simple crowdsourcing platforms (Chen et al., 2014).

5 Considering contemporary researches reveals that almost modeling of every phenomena, natural or manmade, is fronted with kinds of difficulties. Different techniques have been deployed to come over intellectual, reciprocal, technical or mathematical difficulties (Kingsland, 2000).

Nevertheless most of simulator software deploy a mathematical model behind themselves. These models are usually the milestone of whole simulator structure. Designing a proper math model, will significantly optimize the operation of whole 10 simulator. But the problem is the conceptual distance between environment and mathematical space.

A mathematical framework, is a set of basic rules, assumptions and tools, that facilitates solving some kinds of mathematical problems. Customized mathematical framework, logically would be a suitable solution to cover the conceptual gap and empower designing process with proper mathematical tools, techniques and approaches. It simplifies development of software modules due to fast conversion of math functions at the basis of programming codes. Likewise using the 15 framework, one could program main objects' behavior of space directly based on predefined math functions.

In addition, describing an inner structure of software system that simulates a natural or manmade space, is to some extent limited. For instance, UML indicates objects, their properties, behaviors and also object dependencies (Ford, 2005). A programmer is able to understand how to call an object or execute a specific behavior by looking at component object model map. But he couldn't understand how the behaviors act. In the other words, inner structure of behavior includes 20 mathematical procedures that is hidden and there is few standard tools to illustrate it. Customized mathematical framework would mathematically, indicates structure, behind the written program codes.

One similar mathematical framework to ours, has been used by Wei Liu et al. (Liu et al., 2011) to compare protein structures to detect evolutionary relationship among proteins, predicting protein functions and predicting protein structures. They have developed a mathematical framework for protein structure comparison by treating protein structures as three-dimensional 25 curves. Using an elastic Riemannian metric on spaces of curves, geodesic distance, a proper distance on spaces of curves, can be computed for any two protein structures.

Mavandadi et al. (Mavandadi et al., 2012) proposed a methodology for digitally fusing diagnostic decisions made by multiple medical experts in order to improve accuracy of diagnosis. They proposed a probabilistic algorithm as a three component mixture model and solving for the underlying parameters using the Expectation Maximization algorithm.

30 Some have used mathematical frameworks to select features (Genzel and Kutyniok, 2016). They have studied challenge of feature selection based on a relatively small collection of sample pairs f(xi, yi). The observations yi are thereby supposed to follow a noisy single-index model, depending on a certain set of signal variables. A major difficulty is that these variables





usually can't be observed directly, but rather arise as hidden factors in the actual data vectors xi 2 Rd (feature variables). They have proved that a successful variable selection is still possible due to their mathematical framework even when the applied estimator does not have any knowledge of the underlying model parameters and only takes the "raw" samples f(xi, yi) as input.

Cheng has proposed a mathematical framework for spatial decision support system (Cheng, 2017). He has tried to model interdependency of critical infrastructure during geo-disasters. He has quantified interdependency among infrastructure via his framework. Framework works on an asymmetric relation matrix constructed in a bottom-up approach for modeling and analyzing interdependencies of critical infrastructures.

Mathematical frameworks would be used in representation of feature geometry in GIS. Point objects represent entities in

most facility location problems. This can cause solution error and limit the range of potential solutions. Miller has provided a mathematical framework in GIS space for realizing point representation (Harvey and Miller, 2015). He has offered some algorithm to implement his idea.

Considering proposed framework reveals that all facilitate a computational purpose but framework itself doesn't contain all necessary parts. Most frameworks do not contain definitions, principals, proven subjects and toolsets.

Furthermore it is hard to develop them based on themselves or even other things. They rarely would be reused and are difficult to be a part of dynamic developing structure.

### 3.   Mathematical framework for spatial crowdsourcing services

Here, we intended to propose a mathematical framework to manage spatial crowdsourcing services among critical situations

and rescuers through field. Framework mainly utilizes tensors in vector space, differential geometry and transformation of those vectors in a multi-dimensional space.

Assume there is a multi-dimensional and non-orthogonal space for crowdsourcing. Human life requirements are the main dimensions of our proposed space. Air, water, food, shelter, cloth, movement, hygiene, peace and mental health are among the most important and urgent life requirements Figure1. Whenever an incident or any crisis with unpleasant consequences

occurs to any situation, existence or quality of important human related parameters would be threaten. Although there are many indexes to determine life quality that mainly focuses on mental well-being (Barcaccia, 2013) but on disaster or crisis time, it is directly determined based on mentioned parameters (Liam et al., 2012).

**Figure1**

We consider four major classes of objects including the injured, rescuers, resources and forces as the basis of that space. Any

space position would be a tensor of rank 1. These tensors point to the position of crowdsourcing main target objects and reveal their general status according to space dimensions.  Origin status of objects are determined by scalars or tensors of rank 0. Scalars are in fact the quantified status of objects in all space dimensions. Although it seems too difficult to state



individual's properties such as health condition in numeric, but different quantification methods (Quantification (science), 2016) would be deployed to do it.

Framework also includes high-level developed mathematical tools based on specific definitions. Mathematical tools are in form of tensors that acts on state vector of objects. These tools simulate spatial crowdsourcing services and their interactions with four major classes in space. Services conduct crowds through emergency condition via primary field cognition, allocating proper rescuers to injured points, transmitting resources to required points or carrying injured to field hospital or conducting homeless to accommodation points. Framework also contains math tools which measure fatigue of resources and services time-lines.

The only important thing that significantly controls the framework operation is quantification. In social sciences, quantification data is earned due to empirical observation and psychological experiments (Crosby, 1996). These data is then analyzed statistically to form quantified outputs. For intangible properties one would ask others to rate it as a specified scale or designed index (Hong and Sungook, 2004). For unobservable variables it would be replaced by another parameter which is highly correlated. In some cases linguistic quantifier would be deployed (Wiese and Heike, 2003).

To produce scalars which indicates the status of objects in crisis field, two approaches were deployed. Some scalars such as air quality, water quality, food quality, hygiene quality, would be referred to international indexes (Global Standards, 2016). Other group of scalars refer to people properties which would not be quantified simply such as mental hygiene. For these scalars a common form of hedge, other than purely linguistic, that is intuitive (subjective) was set. It is based on probabilities that individuals use in making judgments under uncertainty (Tversky and Kahneman, 1974).

**Figure 1**

As it is illustrated in Figure 1), according to real and required process in crisis filed, different operations and tools shall be discussed. First we will describe framework space and then we will introduce mathematical tools of the proposed framework which manage spatial crowdsourcing services through emergency aiding process.

**3-1 Mathematical Modeling for crowdsourcing emergency responses**

Based on our proposed ontology, main classes would be people who need help or injured, rescuers or people who will help, Resources and Forces. Class which is titled Resources refers to any things which serve the first two classes such as trucks, shelters, foods, water, drugs and so on. The class is titled Forces, dedicates to those which create unpleasant events such as an earthquake or a flood.

Although these classes are indeed in different taxonomy, but all could have common properties such as position, time, life signals, air, water, food supply volume and degree of freedom. Of course there would be other properties which are ignored for the sake of simplicity. Mentioned properties compose the main dimensions of our proposed space. Thus for any given


object belongs to our main classes we assume a vector $v^i$ as contravariant vector (Feng, 2010) as **Error! Reference source not found.**).

$$V_{rescuer}(x, y, t, d, s, a, w, f, sh) = \{r_i\vec{x} + r_{i+1}\vec{y} + r_{i+2}\vec{t} + r_{i+3}\vec{d} + r_{i+4}\vec{s} + r_{i+5}\vec{a} + r_{i+6}\vec{w} + r_{i+7}\vec{f} + r_{i+8}\overrightarrow{sh}|r_i \in R\} \qquad (1)$$

According to, Equation.1 $\vec{x}$ and $\vec{y}$ determine geometric position, $\vec{t}$ stands for time, $\vec{d}$ describes freedom degree of movement

in space that for a captured or closed object or situation it equals to 0, $\vec{s}$ measures life signal, $\vec{a}, \vec{w}, \vec{f}$ stand for air, water and

food supply volume and finally $\overrightarrow{sh}$ determines shelter availability. While these are main vectors of our proposed space and have direction, their coefficients are scalars which define magnitude of main vectors.

### 3.1.1 Crisis occurrence

We can define different operations among tensors of rank1 (Goetz, 1970). For example assume a normal person in space

with a status vector "a" is added by disaster force vector "b", produces a transition vector of "c" that gives new injured

person status in space as in Fig. (3).

**Figure 3**

It simply means that

$$\vec{V}_{injured} = \vec{V_1} + \vec{V}_{force} \qquad (2)$$

Thus, having the filed risk map and hazard magnitude, we will be able to estimate field condition right after hazard by adding force vector to individual residents' vectors in risk area.

### 3.1.2 Corporation among rescuers as multiplication to services

Any objects in our space, would be selectively multiplied in some dimension of space. For an instance we would indicate a

single rescuer in space using Eq. (3):

$$\vec{V}_{rescuer} = r_i\vec{x} + r_{i+1}\vec{y} + r_{i+2}\vec{t} + r_{i+3}\vec{d} + r_{i+4}\vec{s} + r_{i+5}\vec{a} + r_{i+6}\vec{w} + r_{i+7}\vec{f} + r_{i+8}\overrightarrow{sh} \qquad (3)$$

Then if a single rescuer can raise scalar of dimension d, one unit in one unit of time, then two rescuers would do the same in the half of the defined time. This would be indicated as below:

$$I_{m \times m} \quad then\ we\ assume \quad I(3,3) = 0.5\ stating\ time\ create\ I_{m \times m}^n \quad then\ I_{m \times m}^n \times \vec{V}_{rescuer\ m \times m} \qquad (4)$$

Note that a matrix acts on a vector, then result should be a vector. According to Einstein summation convention it could be written as Eg. (5) (Feng, J.C., 2010):

$$M_j^i v^i = \lambda v^i \qquad (5)$$

In this equation, $M_j^i$ represents matrix, $v^i$ stands for vector and $\lambda$ is a coefficient.

### 3.1.2 Crowdsourcing service element composer

In so many situations, a rescuer needs some kind of resources to aid an injured person. For example one would take some food and carry it to injured one. In these situations, there would be some available resources and rescuers in crisis field



waiting for a decision. In order to select suitable pair of resources and rescuers different scenarios are on the table. If time of aiding process is the only important parameter, therefore following solution simply gives suitable output.

$$\left[\left(\begin{bmatrix}1 & \cdots & 0 \\ \vdots & \ddots & \vdots \\ 0 & \cdots & 0\end{bmatrix}_{9\times9} \times \begin{bmatrix}r_i \\ \vdots \\ r_{i+9}\end{bmatrix}^{Rescuer}_{9\times1}\right) - \left(\begin{bmatrix}1 & \cdots & 0 \\ \vdots & \ddots & \vdots \\ 0 & \cdots & 0\end{bmatrix}_{9\times9} \times \begin{bmatrix}e_i \\ \vdots \\ e_{i+9}\end{bmatrix}^{Resource}_{9\times1}\right)\right] \cdot \left[\left(\begin{bmatrix}1 & \cdots & 0 \\ \vdots & \ddots & \vdots \\ 0 & \cdots & 0\end{bmatrix}_{9\times9} \times \begin{bmatrix}r_i \\ \vdots \\ r_{i+9}\end{bmatrix}^{Rescuer}_{9\times1}\right) - \left(\begin{bmatrix}1 & \cdots & 0 \\ \vdots & \ddots & \vdots \\ 0 & \cdots & 0\end{bmatrix}_{9\times9} \times \begin{bmatrix}e_i \\ \vdots \\ e_{i+9}\end{bmatrix}^{Resource}_{9\times1}\right)\right] \qquad (6)$$

Above expression gives dot product of two main vector differences. Diagonal matrix, has only two 1 value and all other values are 0. The result is a scalar which is used to compare different pairs suitability. Dividing the result by time, produces a new scalar that represent suitability volume.

### 3.1.3 Identity element for crowdsourcing services

Although a vector of rescuer, include different dimensions, but it doesn't deploy all its dimensions anywhere anytime. For example when a rescuer move across the XY plate, set its position on proper location or carry something to somewhere, the only available coefficients in the rescuer vector would be $\vec{x}$, $\vec{y}$ and time.

$$\vec{V}_{rescuer} = r_i\vec{x} + r_{i+1}\vec{y} + r_{i+2}\vec{t} \qquad (7)$$

Sometimes different rescuers affect an individual or object subsequently, one after the other. In this case, rescuers' vectors, affect aiding applicant vectors in the same order. Of course, according to limitation of time and space, there would be some temporal gaps. It has been indicated in Figure .

Figure (4)

Temporal gaps because of arrival waiting time, would be modeled via neutralized vector in vector addition operation. However this vector just raises the time. In more accurate situation, this vector would contain other negative parameters such as life signal. We assume it as in Eq. (8):

$$\vec{I}^t = r_{i+2}\vec{t} - (r_{i+2}) \times d_s\vec{s} \qquad (8)$$

In above equation, ds is referred to (deterioration of injured signal life/ time). Figure.4 displays temporal vectors in time-space. Time has been extruded instead of z in Fig. (5):

Figure (5)

### 3.2 Mathematical modeling of Crowdsourcing services

To remove temporal delay time, it is possible to deploy two or more rescuers simultaneously via our services. Furthermore a rescuer would use some kind of resources to aid an injured. In such cases it is not possible to add their vectors. Likewise it would not possible to use transformation matrices whereas they are deployed in a homogeneous space. In our proposed space, dimensions intrinsically are different. Therefore we would have a heterogeneous space.

Thus a combination of two vectors is used. That would be a normal product nor dot or vector product. A normal product of two vectors indeed form a tensor of rank 2.

$$\vec{V}_{rescuer} = r_i\vec{x} + r_{i+1}\vec{y} + r_{i+2}\vec{t} + r_{i+3}\vec{d} + r_{i+4}\vec{s} + r_{i+5}\vec{a} + r_{i+6}\vec{w} + r_{i+7}\vec{f} + r_{i+8}\vec{sh} \qquad (9)$$

$$\vec{V}_{resource} = e_i\vec{x} + e_{i+1}\vec{y} + e_{i+2}\vec{t} + e_{i+3}\vec{d} + e_{i+4}\vec{s} + e_{i+5}\vec{a} + e_{i+6}\vec{w} + e_{i+7}\vec{f} + e_{i+8}\vec{sh} \qquad (10)$$





$$\vec{V}_{rescuer} \times \vec{V}_{resource} = r_i e_i \vec{x}\vec{x} + r_i e_{i+1} \vec{x}\vec{y} + r_i e_{i+2} \vec{x}\vec{t} + r_i e_{i+3} \vec{x}\vec{d} + r_i e_{i+4} \vec{x}\vec{s} + \cdots \tag{11}$$

Result will be a matrix as indicated in Eq. (12):

$$A_{9\times9}^{Trans} = \begin{pmatrix} r_i e_i & \cdots & r_i e_{i+8} \\ \vdots & \ddots & \vdots \\ r_{i+8} e_i & \cdots & r_{i+8} e_{i+8} \end{pmatrix} \tag{12}$$

## 3.3 Crowdsourcing service strength

Combination of different resources and rescuers produce a new vector with certain strength. In order to measure magnitude of this vector following solution would be used. Note that here geometrical distance is ignored.

$$\left[ \left( \begin{bmatrix} 0 & \cdots & 0 \\ \vdots & \ddots & \vdots \\ 0 & \cdots & 1 \end{bmatrix}_{9\times9} \times \begin{bmatrix} r_i \\ \vdots \\ r_{i+9} \end{bmatrix}_{9\times1}^{Rescuer} \right) \cdot \left( \begin{bmatrix} 0 & \cdots & 0 \\ \vdots & \ddots & \vdots \\ 0 & \cdots & 1 \end{bmatrix}_{9\times9} \times \begin{bmatrix} e_i \\ \vdots \\ e_{i+9} \end{bmatrix}_{9\times1}^{Resource} \right) \right] \tag{13}$$

In above matrices, diagonal elements are 1 expect for primary 3 first ones that are 0. This tool would be used to measure suitability of different possible pairs of rescuers and resources.

### 3.4 Length of curve as crowdsourcing service cost

An injured status point, would travel through space according to the time. Therefore tracking different positions of injured state vector, introducing a curve in space. A state vector length of curve is defined (O'Neill, 1966-1997) in Eq. (14):

$$lc = \sum_{i=1}^{n} |r(t_i) - r(t_{i-1})| \tag{14}$$

Where $lc$ is length of curve and $r(t_i)$ represents state vector of time $t_i$. Now, Assuming r(t) as state vector length of curve in Eq. (15):

$$lc = \left[ \sum_{t=1}^{n} \sqrt{ \left( \begin{bmatrix} i_t^1 \\ \vdots \\ i_t^9 \end{bmatrix}_{9\times1}^{injured} - \begin{bmatrix} i_{t-1}^1 \\ \vdots \\ i_{t-1}^9 \end{bmatrix}_{9\times1}^{injured} \right) \cdot \left( \begin{bmatrix} i_t^1 \\ \vdots \\ i_t^9 \end{bmatrix}_{9\times1}^{injured} - \begin{bmatrix} i_{t-1}^1 \\ \vdots \\ i_{t-1}^9 \end{bmatrix}_{9\times1}^{injured} \right) } \right] \tag{15}$$

Length of curve, here has slightly different meaning in our proposed framework. This would be interpreted as cost of overcoming a task or a mission. This depreciates rescuer limited resources. In the other words, rescuers have not unlimited potential and according to their different resources can continue aiding process till specific time of day. Lack of resources, force them to stop. Although construction of precise criteria to measure all possible resource of rescuers would seem too difficult, but proposed solution as the length of curve, gives a simple clue to measure consumption of rescuer resources. Thus having a certain volume for any given rescuer, we would be able to determine how long he could resist in crisis field as in Eq. (16).

$$resistance_i = \int_0^t lc(t) dt \tag{16}$$



## 4    Evaluation

We compared our proposed mathematical framework with the prevalent mathematical frameworks. It would present more aspects of framework properties and behaviors and would help us for further develops and refinements. To do the evaluation, three groups of master student including 12 in GIS, 7 in mathematics and 10 in computer, were presented with framework
operation and abilities. Then they were requested to fill following table. Average of results has been expressed in Tab. (1).

**Table (1)**

Furthermore it is possible to check framework results against real situation. For an instance, based on earthquake report of Bam which is located in southeast of Iran, temporal movement of a rescuer was implemented in real space. Then it was compared with the plan which has been conducted by our proposed framework. Real surveyed path has been indicated in
yellow color in Fig. (6).

Figure (5 )

As it is clearly seen that a considerable shift is occurred among two graphs. Such shift is mainly produced by unsuitable selection of rescuers for proper missions. This temporal shift endangers people life via extending emergency time.

Further investigations were made due to simulating an aiding scenario in SIMIO simulation software. Scenario was run by 4
rescuers with different skills and 4 injured persons in SIMIO and our proposed framework. Then according to the time, location of rescuer in both environment were compared and its distance was extracted Fig. 7(a).  Results revealed that approximately for the first 2 hours, for carrier and medical rescuers both behave same, but for the next hours, SIMIO departs from our framework. But for debris remover and rescuer who is responsible to amenities, both behave in a same manner. Figure 7(b) depicts that position vector of different injured persons in multi-dimensional space after 3 hours depart from
SIMIO results. This stresses constraint points in approximate 4 hour spans, to prevent significant deviation. In other test, number of state vector of all features, including rescuers, resources and injured persons, were increased and then distance between framework and SIMIO results were recorded. This comparison Fig.7(c) revealed that difference is gradually descended by increasing number of features. This would be direct result of theory of errors.

Figure 7

Combination of rescuers and resources is done via normal product of their state vectors. Measurement of framework accuracy results against simulated values Fig.7 (d), indicates that error volume is approximately constant during a time span of 8 hours and is relatively less than 10%. Other tool in framework expresses fatigue volume of rescuers in form of a first rank tensor. According to designed index in SIMIO, it expresses acceptable error at least for the first 3.5 hours in an 8-hour time span fig7.(e).  Framework also selects rescuers for different situations based on their skills, positions and powers. 3
different rescuers were taken in a scenario and continuously were selected by framework tools 8 times. These selections were compared against simulated scenario. Results revealed relative error of 3/8 for carrier rescuer, 1/8 for medical and 0 for debris remover fig7.(f).



Altogether, tests indicated acceptable consistency according to SIMIO environment for short time span. In the other words, in order to calibrate math framework tools, they should be tied with boundary constraint numbers and mid control points.

## 5 Conclusion

In this paper, a mathematical framework was presented. This framework aids the rescuing process of a crisis field via spatial
crowdsourcing services. To construct framework, a multidimensional vector space was defined. Then we projected real environment of a crowdsourcing field to our space. This mainly has used the basic concepts of differential geometry in vectors. Using different topics in differential geometry such as length of curve or curvature, simplified defining specific tools for our purpose.

Critical objects were simulated due to vectors or tensors of rank1. Then to construct interactions among the objects and
crowdsourcing services, higher rank of tensors e.g. tensors of ranks 2, dot product of vectors, normal product and some other combination operators, as our new specific tools, were designed and utilized.

These specific tools beside a few definitions, constructed our new proposed framework. Framework is then able to simulate different crisis field object behaviors and our crowdsourcing services in space. Tools give the ability to track objects' movement in all dimension of space including time. Upon this ability, we would be able to form influence-ability constrain
with more dimensions comparing it in time geography issue.

Furthermore it could let us to document main environment objects and interactions as well as crowdsourcing services. Such documentation style could fully describe internal details of behavior to developers. It would also simplify programming process. Likewise tools in our proposed framework enable us to examine objects and their behaviors just on piece of paper without requiring simulation software.

Although mathematical tools in our framework are simple to use but the main problem arises from quantification. This subject is critically important especially when we decide to deploy framework as simulator system. Objects in a crisis field are in real space so they would have properties that almost are in non-numerical nature. Regardless of several optimized quantification methods, uncertainty level is still high.

In addition framework should act on space which contains different dimensions with different nature. So it would be too
difficult to regulate suitable scale among these dimensions. In other words, incorporated actions among objects in space would be deviated.

Designing more accurate models in form of functional vectors and regulating those vectors by real values in environment would ameliorate partial accuracy but generally it remains crucial. So in this subject, it seems some new effective solution is required.

Nevertheless it also seems that customizing mathematical software environment such as Matlab, beside other ancillary tools in differential geometry and tensors, would improve efficiency of framework.




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





## Figures

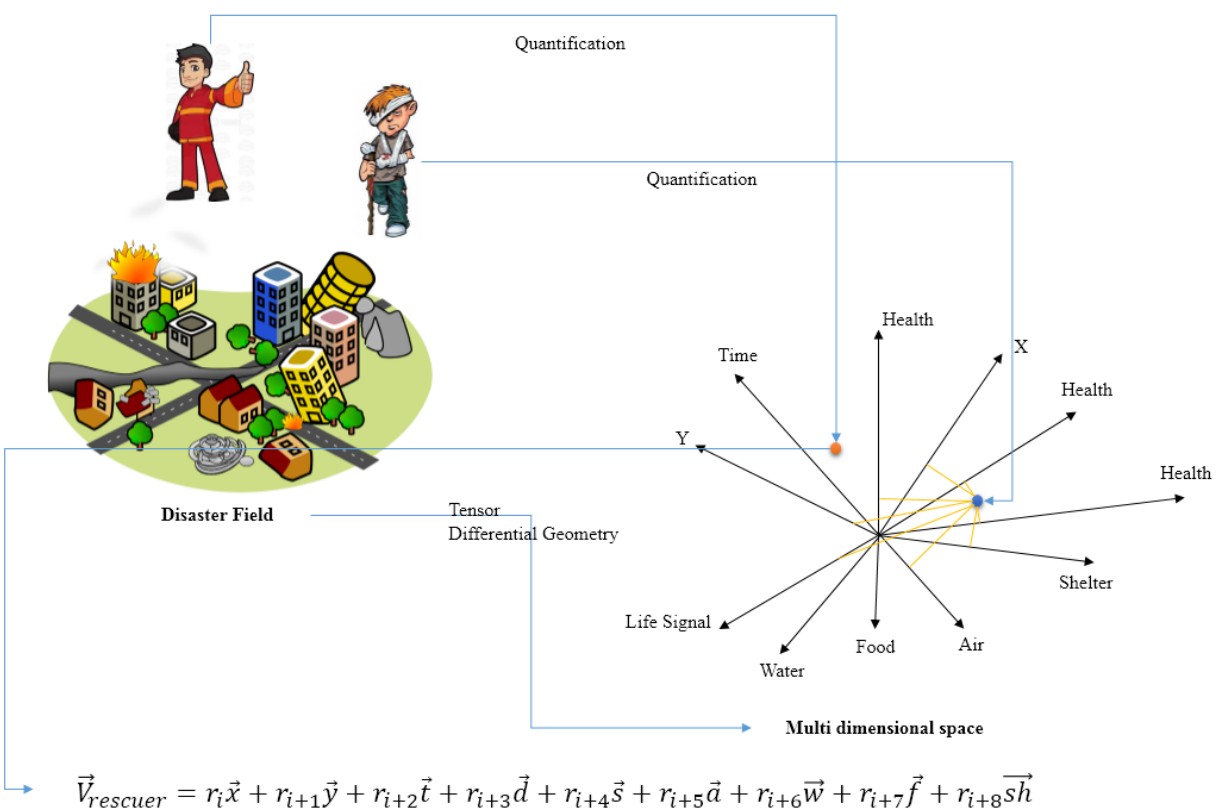

$$\vec{V}_{rescuer} = r_i\vec{x} + r_{i+1}\vec{y} + r_{i+2}\vec{t} + r_{i+3}\vec{d} + r_{i+4}\vec{s} + r_{i+5}\vec{a} + r_{i+6}\vec{w} + r_{i+7}\vec{f} + r_{i+8}\overrightarrow{sh}$$

**Figure 1: Construction of multi-dimensional space and projecting crisis objects in the abstract space. $\vec{x}$ and $\vec{y}$ determine geometric position, $\vec{t}$ stands for time, $\vec{d}$ describes degree of freedom in space, $\vec{s}$ measures life signal, $\vec{a}, \vec{w}, \vec{f}$ stand for air, water and food supply volume and finally $\overrightarrow{sh}$ determines shelter availability**



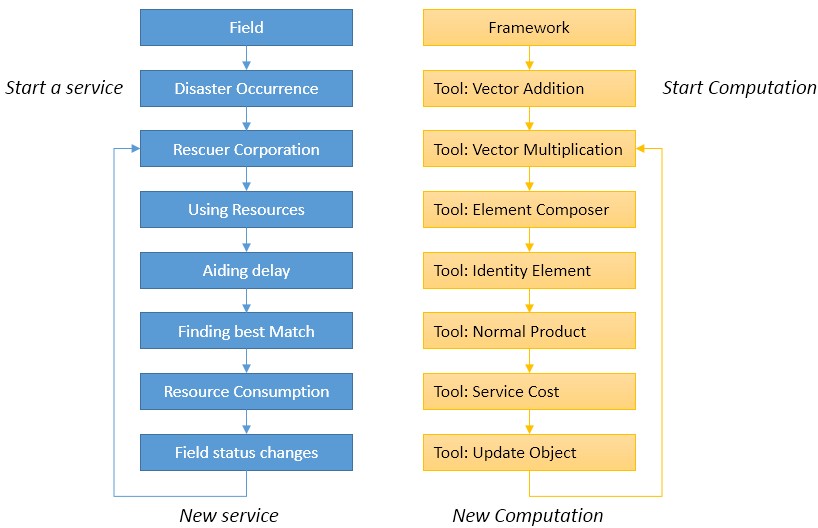

**Figure 2: Recovery of crisis field status using framework tools**

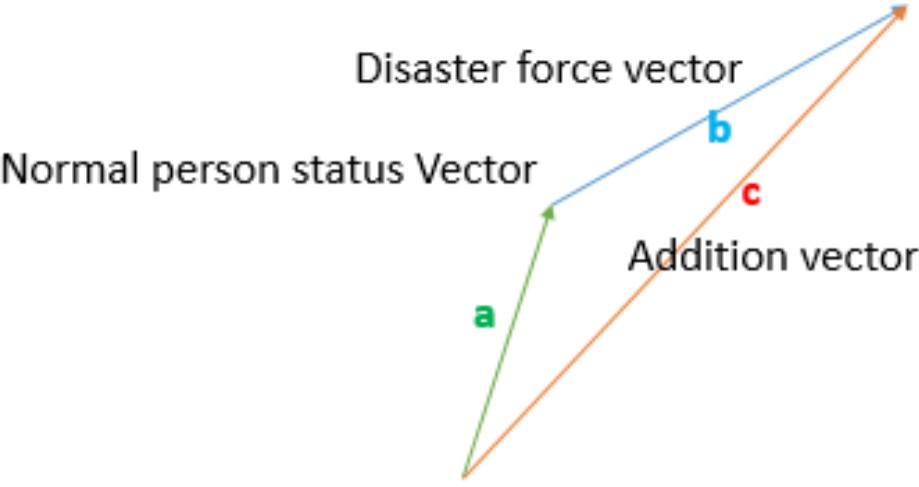

**Figure 3: Addition vector of primary status of a person and disaster force vector**



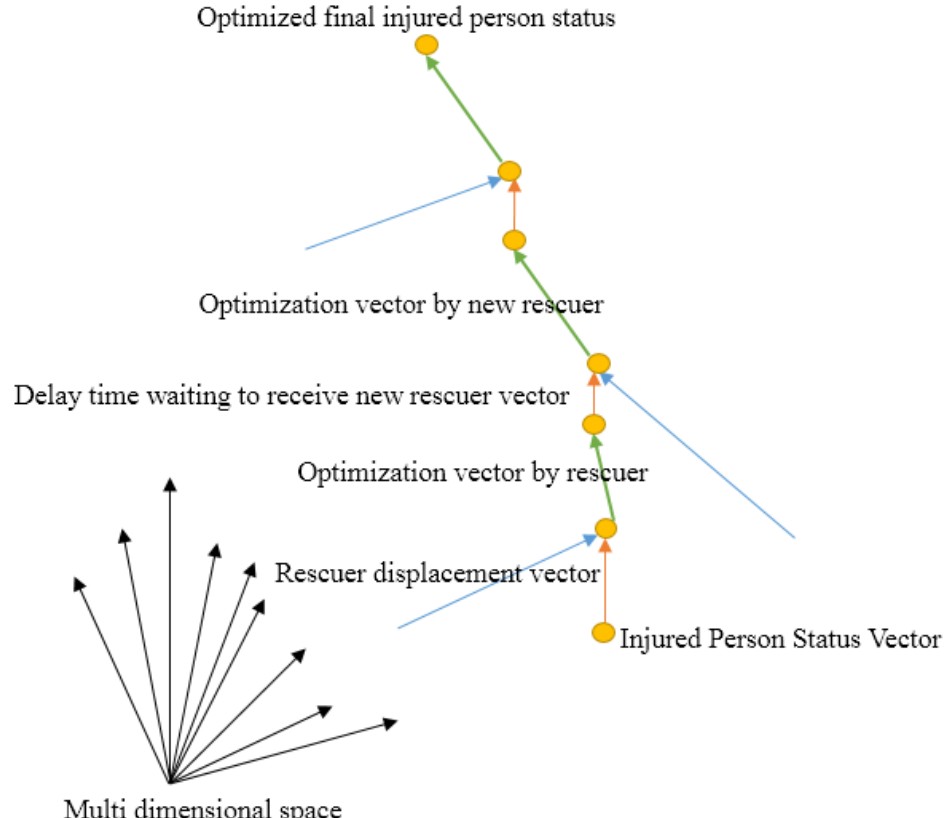

**Figure 4: Temporal gaps among services as delay time in figure**





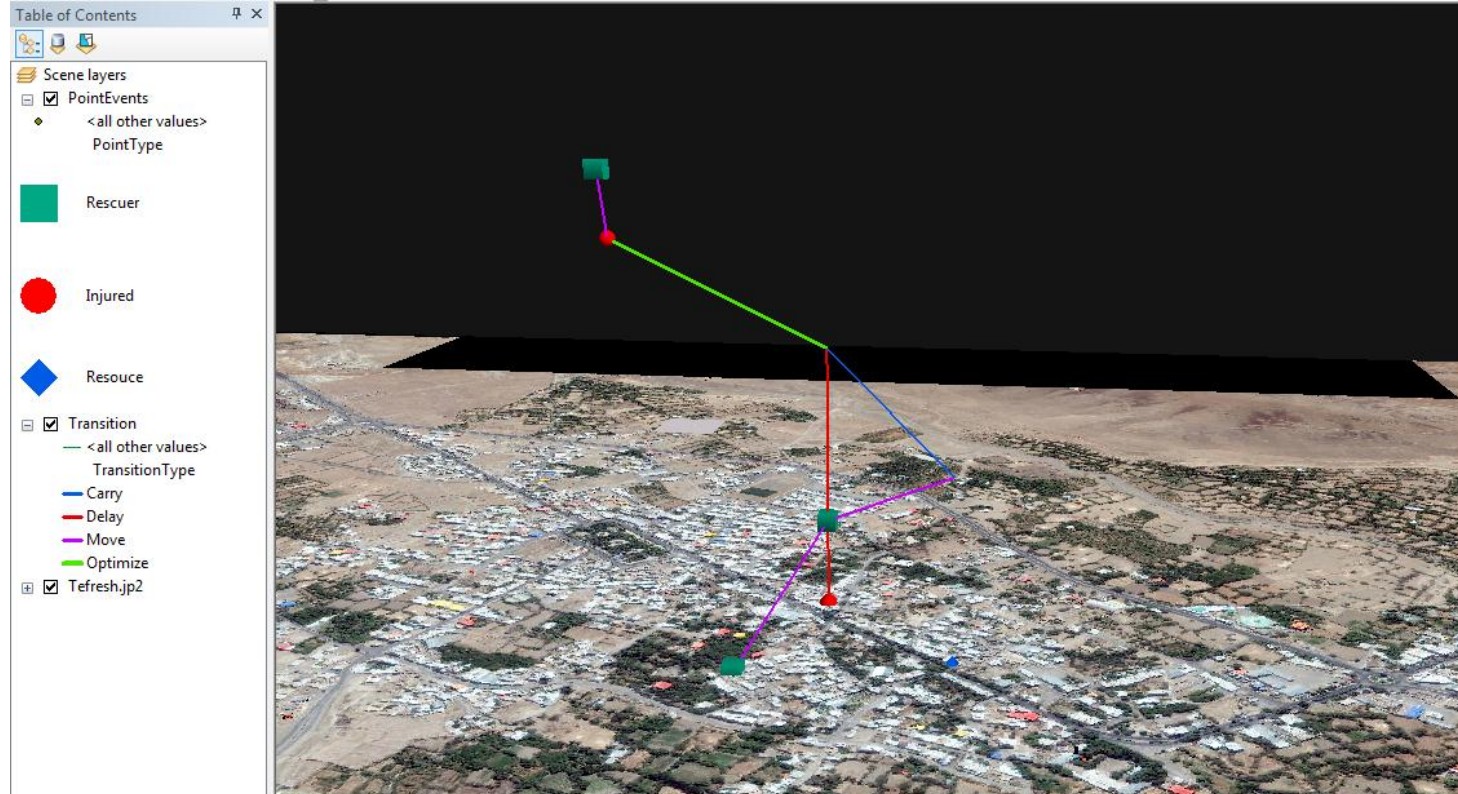

**Figure 5: 3D vectors in time-space environment where vertical line indicates time axis**

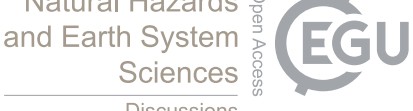

**Figure 6: Recommended path in multi-colored graph, each color for different rescuer category and real survived yellow path for a rescuer in Bam's earthquake**





**Figure 7: (a) Two dimensional difference between simulated rescuers' position and framework results based on time axel, (b) Multi-dimensional distance between simulated positions of injured persons and framework estimations based on time, (c) Position difference of all main feature classes by increasing number of features, (d) Normal product differences of two cases; corporation and usage normal products based on time of service, (e) Difference between simulated fatigue volume of rescuers and framework results by increasing time, (f) framework selection status regarding three major group of rescuers by comparison to simulated environment**



**Tables**

| Row | Framework Specifications | Matlab | Single Purpose Mathematical Framework | Crowdsourcing Framework | Visio | Computer Framework |
|---|---|---|---|---|---|---|
| 1 | Extendibility | 10 | 1 | 7 | 5 | 10 |
| 2 | Comprehensivability | 8 | 3 | 5 | 9 | 6 |
| 3 | Completeness | 10 | 8 | 7 | 3 | 8 |
| 4 | Work speed | 7 | 10 | 8 | 9 | 7 |
| 5 | Ease to use | 4 | 10 | 7 | 9 | 5 |
| 6 | Collectivity | 10 | 1 | 6 | 8 | 9 |
| 7 | Elegance | 9 | 3 | 5 | 6 | 5 |
| 8 | Readability | 9 | 5 | 6 | 10 | 4 |
| 9 | Process describing | 7 | 1 | 8 | 3 | 5 |
| 10 | Structure describing | 2 | 3 | 6 | 7 | 4 |
| 11 | Accuracy | 9 | 5 | 7 | 2 | 8 |
| 12 | Fitness for use | 10 | 4 | 6 | 5 | 10 |
| | | | | | | |
| | Total Score | 95 | 54 | 78 | 76 | 81 |

Table 1: Comparing Frameworks

