# Peer review of "A Mathematical Framework for Crisis Spatial Crowdsourcing Services"

_Natural Hazards and Earth System Sciences, 2018_

## Referee Comment (RC1) · Anonymous Referee #1 · 12 Sep 2018

The proposed Manuscript illustrates an information system aimed at describing and, possibly, managing emergency and recovery procedures after the occurrence of a natural disaster. The framework is supposed to describe each object in the field by a vector mathematical representation as a function of time, and the interactions between them by matrix and tensor algebra.

In principle, the topic is of great relevance and the Abstract describes an interesting approach, though it is clear from the beginning that the project is rather ambitious to be ready for a plug-and-play application, as the Authors seem to suggest both in the Abstract and in the Introduction. The Introduction itself seems to contain pertinent references, even if it could be expanded a bit, in my opinion. I could suggest a few references, mostly related to crowdsourcing for spatial data and natural hazards, exploitation of data from social media, application of scientific advances to the local administrations, and emergency response. I believe these would help putting the Manuscript in a broader perspective but, once again, this could be done in a Manuscript containing relevant results, which is not the case here, in my opinion.

As a matter of fact, I believe that the Manuscript suffers from severe limitations and it is not suitable for publication as a regular paper in NHESS, in the present form. I will briefly describe such limitations in general terms, first, and is some more detail, later on.

* General comments:

One major limitation of the proposed Manuscript is that the text is really difficult to follow, probably due to non-native English Authors. I believe the Authors should seek help from a professional service to make the language more fluent and get rid of the many ambiguities that, in my opinion, can be found in the current form of the Manuscript and that severely hamper readability.

A second, important issue is that the mathematical framework is introduced in a succint way, and it is diffult to judge not only if it represents a correct description of the physical system under investigation, but also whether it is a sound mathematical representation from the very beginning. The objects involved in the framework are introduced without explanation, so that the reader must faithfully believe that each vector can properly describe the quantities it is supposed to represent. Interactions between objects are cast in the form of vector manipulation operations, but such a correspondence is just postulated and never justified or explained. For example (I will give more details in the following), Eq. (2) says that the vector describing the state of an injured person after the occurrence of a disaster can be obtained by summing a vector describing the state before the occurrence, and a "force" vector, describing the effect of the disaster itself.

Where does this "force" vector come from? How can it be obtained from knowledge of the type, location and magnitude of the event triggering a disaster? Even the fact that these quantities can be known at all, which I do not believe can be given for granted.

Eventually, the Authors promised in the Abstract and Introduction that the proposed framework would enable management of volunteered citizen resources, but results and conclusions reported in the Manuscript do not actually support such a statement. The presented results are, at best, a description of the displacements of objects in the field, but I could not find any attempt of proper management, and/or optimization, of resources and volunteers. Aslo, the supposedly successful comparison of results obtained within the proposed framework and existing software and real field data, is very poor and no objective measure is given about the outcome of the comparison. Actually, no description of what happened in the real-case scenario was given.

A full solution of all the three issues mentioned above is mandatory, in my opinion, to consider the Manuscript for publication in a scientific journal.

* Specific comments:

In the Abstract, it is stated that "The framework inherits tensor and differential geometry specifications". I did not actually find enough evidence of operations borrowed from differential geometry, as no integral nor differential calculus is involved in the presented framework. I do not believe that the elementary linear algebra operations described in the Manuscript can be classifed as differential geometry.

In the Introduction, the Authors point out that often the maths hidden behind existing software is unknown, and this is a limitation when it comes to simulation of natural events. Which is probably true. Nevertheless, this holds true for any commercial software and giving examples in very different research fields (protein structure and medical diagnoses) is useless. Also, in line 30, page 4, the Authors refer to "features" without explaining what they actually refer to, and I got lost in the following description.

Interactive
comment

In Section 3, line 22, page 5: the statement "Assume there is a multi-dimensional and non-othogonal space for crowdsoucing" contains a few problems, in relation to what follows. First, the dimensions listed right after that sentence: are these the dimensions of all the following vectorial quantities? This is never elucidated. Also, the listed "dimensions" are said to be "among" the most important and urgent and life requirements; what are the remaining dimension, then? Later on, from lines 29, page 5 to line 2, page 6, the description of the active objects in the framework relies on the "space dimensions" again, with no further explanation. Moreover, the "origin status" of objects are determined by scalars. What does it mean? Such objects were described as vectors a few lines above; how can they be a scalar at one time, and a vector at a later time? Eventually, it is stated that "quantification methods would be deployed", but I don't actually understand where this is done. In conclusion, this Section does not contain the necessary information to understand what comes next.

In Section 3.1, the many equations describing the framework are impossible to understand, in my opinion. Equation (1) seems to define the status vector of a "rescuer" as a function of different quantities. In particular, such quantities are vectors themselves, and they must be summed to obtain the "rescuer" vector status. Why are x, y, t, d, s, a, w, f, sh vectors themselves? In what space are they defined? Are they homogeneous quantities, so that the sum of them is actually well-defined? I really do not understand. Is time a vector?

Equation (2) was already discussed above.

Equation (3) seems to be the same as Eq. (1). What is the difference, and why?

From Eq. (4), a whole new set of problems arises: the Authors define matrices acting on some object state vector, to evolve its status from some time t to a later time. How do these matrices come from? What are the physical laws encoded in the linear operators, here mathematically represented by matrices? There is no indication whatsoever about this crucial point here. In absence of such a description, the whole framework looks

like a game in which one can plug in any V_force or interaction matrix and check the outcome, without knowing if it is reasonable or not. The same comments apply to Eqs. (6) and (13).

In Section 3.2, lines 24 to 29, page 8, it is stated that status vectors of two different rescuers would live in different spaces, and they cannot be combined/added. Why are the spaces different? Eq. (11) describes a vector operation which is not a dot nor a vector product. What does this vector operation mean, where do the values of the coefficients come from, and what does the resulting matrix represent? This is totally obscure to me.

In Section 3.4, I do not understand the relationship between Eqs. (15), (16). Time dependence in Eq. (15) is given by the values of the involved quantities at the available time steps, $t\_i$, $i=1, .., n$. thus, it looks that no finer granularity in time is defined. How can one integrate over dt, as in Eq. (16)? Is looks like there is not time dependence left in lc, after $\sum^n_{t=1}$, so lc(t) seems to actually be indepedent of time. Maybe I did not understand the notations, here.

In Section 4, the results are "checked against a real situation". First of all, I can find no description of the real situation in the text, other then "temporal movement of a rescuer was implemented in real space" (line 8, page 10). Assuming that such data was actually available, it just provides information about the spatial location of a "rescuer". How about all the other dimensions of the state vectors? Where is such data, how was it obtained, how was it encoded in the state vectors?

Starting from line 7, page 10, a comparison is performed between the results of the proposed framework and the "real situation". Such comparision consisted in "visually comparing" the curves in Figs. 5 and 6. I see two problems with this procedure: a) I believe that a visual comparision is nowhere near a meaninguful comparison of two different numerical quantities; and most importantly, b) this result looks like a (time-depedent?) localization in time of a "rescuer", which is again nowhere near the management of

resources during a crisis. I understand, as I mentioned in the "general comments", that this objective would be an ambitious one, which can be obtained through partial steps, but still the results obtained in this Manuscript (if any, given that a visual comparison is not good enough, in my opinion) seem to be very different from the original claim.

Line 14, page 10: the Authors mention the SIMIO simulation software, which could be anything, since it is never described. Morevoer, the results of the proposed framework and the newly introduced software are compared in a visual way, again, so: same comments as above.

Line 25, page 10: "combination of rescuers resources is done via normal product of their state vectors". This operation, again, is not epxlained and looks just arbitrary. Consequently, the meaining of its results is impossible to understand.

The Conclusions section does not help in solving the issues listed above. As a matter of fact, the Authors state that "this framework aids the rescuing process of a crisis field via spatial crowdsoucring services", which is not supported by the content of the Manuscript. In line 9, page 11, they say that "to construct interactions among the objects and crowdsourcing services, tensors of rank 2, dot product of vectors, normal product and some other combination operator, as our new specific tools, were designed and utilized". This sentence does not contribute in elucidating HOW the interactions were built, and what each class of vector manipualation mean, in the proposed framework. This is particularly critical in view of the fact that the Authors complained about the lack of description, in the majority of existing softwares, of behind-the-scenes maths. Providing an obscure or unclear description of mathematical operations is probably worse than hiding such an explanation. In lines 21-22, page 11, it is stated that "objects are in real space so they would have properties that almost are in non-numerical nature". How did the Authors cope with this issue? This is not clear at all, since in the real case scanario adopted as a test case the available data was not described, and they way it was ingested into the proposed framework was not described either.

The reference Section contains a few entries that are questionable, for example: Barcaccia, (2013); Gansky, (2015); Howe, (2009).

* Technical comments:

The only use of Fig. 3 seems to be the illustration of the addition of two vectors, which is probably not needed. The same goes for Fig. 4, if no further explanation is provided.

---

## Referee Comment (RC2) · Anonymous Referee #2 · 3 Oct 2018

This research presents a crisis crowdsourcing mathematical framework. I believe that this paper is useful, but the current manuscript is not publishable. It is not well-structured. A lot more efforts are needed to improve the manuscript editing and logic flow.